# Evaluating and Enhancing the Robustness of Code Pre-trained Models through Structure-Aware Adversarial Samples Generation

**Nuo Chen**[1], **Qiushi Sun**[2,3][\*] **Jianing Wang**[1], **Ming Gao**[1], **Xiaoli Li**[2][†] **Xiang Li**[1][†]

[1] East China Normal University, Shanghai, China

[2] Institute for Infocomm Research, A*STAR, Singapore

[3] National University of Singapore, Singapore

nuochen@stu.ecnu.edu.cn, qiushisun@u.nus.edu, lygwjn@gmail.com

xlli@i2r.a-star.edu.sg, {xiangli,mgao}@dase.ecnu.edu.cn

## Abstract

Code pre-trained models (CodePTMs) have significantly advanced the field of neural code intelligence. Despite their capabilities, these models are susceptible to adversarial attacks that subtly modify the model inputs, resulting in incorrect outputs or predictions. Previous methods of robustness evaluation for CodePTMs primarily stem from a textual perspective, without explicitly taking into account the structure of the code. Furthermore, prior studies fail to encompass a broad enough spectrum of tasks and models. In this paper, we propose a set of novel robustness evaluation methods based on the intrinsic structure of the code. Specifically, we first launch adversarial attacks on crucial identifier tokens and sub-tree structures to explore the impact of imperceptible perturbation. Then, we perform global restructuring of the code using different traversal methods for abstract syntax trees, aiming to explore the model's sensitivity to input samples with equivalent information. Moreover, for each scenario, we employ adversarial training methods to explore the possibility of restoring the performance of perturbed models. For both code understanding and generation, our proposed method has demonstrated its effectiveness across a wide range of models and tasks, thereby allowing us to make one step forward in our understanding of the inner mechanisms of CodePTMs. Our codes and data are publicly available at https://github.com/nchen909/CodeRobustness.

## 1 Introduction

Pre-trained language models have revolutionized the landscape of natural language processing (NLP) (Devlin et al., 2019; Radford et al., 2019; Liu et al., 2019; Qiu et al., 2020, *inter alia*). While these transformer-based models (Vaswani et al., 2017) have achieved great success in NLP, their counterparts trained on code (Feng et al., 2020; Guo et al., 2021) have also made remarkable strides in the field of neural code intelligence (Xu and Zhu, 2022; Xu et al., 2022; Zan et al., 2023). Despite their impressive capabilities, CodePTMs still retain prevalent weaknesses inherent in language models: they are sensitive to the input sequence and are susceptible to adversarial attacks. The model's vulnerability to variations in input can impair its generalization (Wang et al., 2022; Baniecki and Biecek, 2023). Adversarial examples, though imperceptible to humans, can deceive CodePTMs into generating incorrect predictions or code sequences in downstream tasks (*e.g.*, clone detection, code summarization). Unlike similar attacks for images, audio, and natural languages, the structured nature of programming languages introduces distinct and novel challenges. While methods have been proposed by researchers as potential countermeasures against attacks (Szegedy et al., 2014; Goodfellow et al., 2015; Jia and Liang, 2017; Kang et al., 2018; Zhou et al., 2021), the construction of adversarial samples remains an ongoing concern. Adversarial training often involves exposing the model to adversarial examples during training so that the model learns to defend itself against such examples when it encounters them in the future. However, for CodePTMs, an approach that incorporates code structure is yet to be substantially established.

In representative works on robustness analysis targeting code scenarios, Yang et al. (2022) pioneer an example generation method that balances both natural semantic and operational semantics. Recently, Jha and Reddy (2023) leverage the structure of code to propose code-specific adversarial samples generation, which can be used to evaluate the vulnerabilities of CodePTMs. While these studies concentrate on generating adversarial code samples, there is an absence of explicit modeling of the structure of code. Furthermore, the existing

---

[\*] Work done while interning at Institute for Infocomm Research, A*STAR.

[†] Equal advising.

analyses remain inadequate in examining different architectures of CodePTM and corresponding downstream tasks, which in turn, hampers the generalizability of the conclusions drawn.

In this paper, we propose a comprehensive framework based on the structural information of the code, which integrates sample generation and adversarial training, aiming to conduct a thorough evaluation of the robustness of CodePTMs.

We first conduct an assessment through the perturbation of model inputs. Specifically, we propose two strategies: (1) Generating adversarial samples that are imperceptible to humans to launch adversarial attacks on the model. (2) Leveraging the syntax of the code, namely, the structural information of AST to reconstruct the input sequence into a new one that preserves equivalent information to probe models' sensitivity to input.

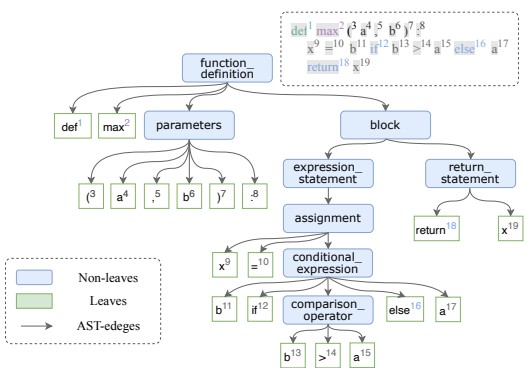

Figure 2: A Python code snippet with its parsed AST.

Then, inspired by adversarial training, we trained our model on samples constructed from code structures, significantly enhancing the model's robustness in the face of structural attacks. Additionally, we conduct an in-depth analysis of the experimental results and validate that the methods to counter various types of attacks are generalizable.

Our contributions can be summarized as follows:

- By perturbing text while maintaining equivalent information, and leveraging adversarial attacks, we unveil both the vulnerabilities and sensitivities of various CodePTMs.

- We utilize a range of adversarial training approaches to recover the performance of disturbed models and conduct an in-depth examination of the unique attributes displayed by different models across a spectrum of tasks.

- Experiments on extensive code-related tasks across different programming languages

demonstrate the effectiveness of our method.

## 2 Related Works

### 2.1 Code Pre-trained Models

Following the success of pre-trained language models (Qiu et al., 2020) in NLP, code pre-trained models (CodePTMs) have recently demonstrated remarkable success in a wide range of downstream tasks in the domain of neural code intelligence. By pretraining on massive code-based data (*e.g.* GitHub repositories), these models can learn rich contextual representations that can be transferred to code-related downstream tasks. Feng et al. (2020) use bimodal data from CodeSearchNet (Husain et al., 2019) to train CodeBERT that shares the same model architecture as RoBERTa (Liu et al., 2019). Then, GraphCodeBERT also uses the same architecture while additionally considering the inherent structure of code, specifically, the data flow graph. There are also models with encoder-decoder architectures, such as CodeT5 (Wang et al., 2021) and PLBART (Ahmad et al., 2021), which inherit the multi-task training strategies of T5 (Raffel et al., 2020) and BART (Lewis et al., 2020). UniX-coder (Guo et al., 2022) harness the UniLM (Dong et al., 2019) architecture and is pre-trained on cross-modal data to support both code understanding and generation. Moreover, decoder-only CodePTMs are crafted to generate high-quality code sequences, which can be utilized for program synthesis (Chen et al., 2021) and even excel in programming competitions (Li et al., 2022).

### 2.2 Adversarial Attacks for Language Models

Despite the remarkable achievements of language models existing literature reveals their susceptibility to adversarial samples, which involve subtle perturbations to initial inputs (Chakraborty et al., 2018). In NLP, this technique was initially employed to evaluate the robustness of models across different tasks (Jia and Liang, 2017; Iyyer et al., 2018; Belinkov and Bisk, 2018, *inter alia*). With the rise of pre-training, BERT-Attack (Li et al., 2020) is first proposed that uses pre-trained language models to generate adversarial samples. For the scenario of code-related tasks, methods for AST-based neural networks was first proposed (Yefet et al., 2020; Zhang et al., 2020). Then, Yang et al. (2022) adversarially transform inputs to make victim CodePTMs produce wrong outputs while considering the natural semantics

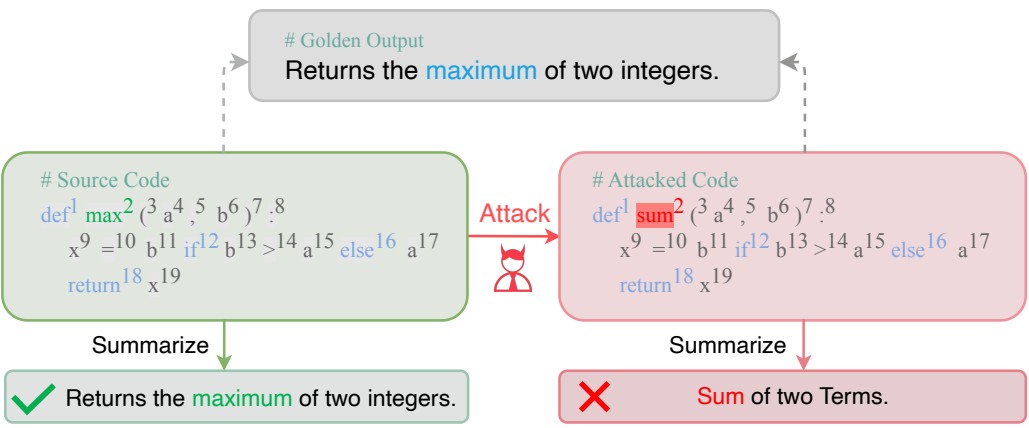

Figure 1: Illustration of curating adversarial samples based on the structure of code to attack CodePTMs. For brevity, we employ the CodeT5 model backbone, showcasing the strategy of signature attack in code summarization tasks using Python code snippets from training sets.

of code. Recently, CodeAttack (Jha and Reddy, 2023) utilizes code structure to generate adversarial samples for evaluating the vulnerabilities of CodePTMs. Zhang et al. (2023) harnesses the uncertainty of CodePTMs' output, utilizing it to guide searching for adversarial examples through variable name substitution. Distinct from these methods, we have adapted various attack strategies to fit the code scenarios. Moreover, we concurrently examine the vulnerabilities of both code generation and understanding tasks for CodePTMs across different architectures.

## 3 Method

### 3.1 Adversarial Attacks on Code Pre-trained Models

**Subtree Attack.** Randomly drop a non-leaf node and its descendants from a code snippet's parsed AST. In the AST, non-leaf nodes typically represent higher-level constructs of the code, such as conditional statements, function definitions, and so forth. Deleting a non-leaf node along with all its child nodes may imply the disruption of a part of the code, such as a loop, a branch of a conditional statement, or a function call. Specifically, as shown in Algorithm 1, a code snippet is first parsed into an AST with all non-leaf nodes identified. Then, one of them is randomly selected and dropped, along with all of its child nodes. Following this, the modified AST is re-converted back into code snippets. In the rare cases where the AST has no non-leaf nodes, the original code snippet is directly returned.

**Signature Attack.** Unlike natural language, the signature of a function often contains more information than other tokens in the sequence. Thus, we

---

**Algorithm 1** Subtree Attack

**Input:** Code snippet $c$
**Output:** Modified code snippet $c'$
1: **procedure** DROPSUBTREE($c$)
2:      $T \leftarrow$ GetAST($c$)
3:      $leaf\_parents \leftarrow$ GetLeafnodesParents($T$)
4:      **if** $leaf\_parents \neq None$ **then**
5:          $parent\_to\_drop \leftarrow$ RandomChoose ($leaf\_parents$)
6:          RemoveChildrens($T, parent\_to\_drop$)
7:          $c' \leftarrow$ ASTtoCode($T$)
8:          **return** $c'$
9:      **else**
10:          **return** $c$
11:      **end if**
12: **end procedure**

---

propose another approach to straightforwardly constructing adversarial samples that involve randomly replacing the signature of an input function with another word from the vocabulary. Although altering the function signature does not change the intrinsic logic of the function, it can modify the code snippets' context. This subtle change could present challenges to CodePTMs that seek to understand code at a semantic level. For instance, suppose we have a function with the signature add(a, b), which is used in the code to perform additional operations. If we change this function's signature to subtract(a, b), the intrinsic logic of the function (performing addition operations) remains unchanged. However, the semantics of the function undergoes a significant transformation. Sequential models would typically expect the subtract function to perform subtraction operations, not addition.

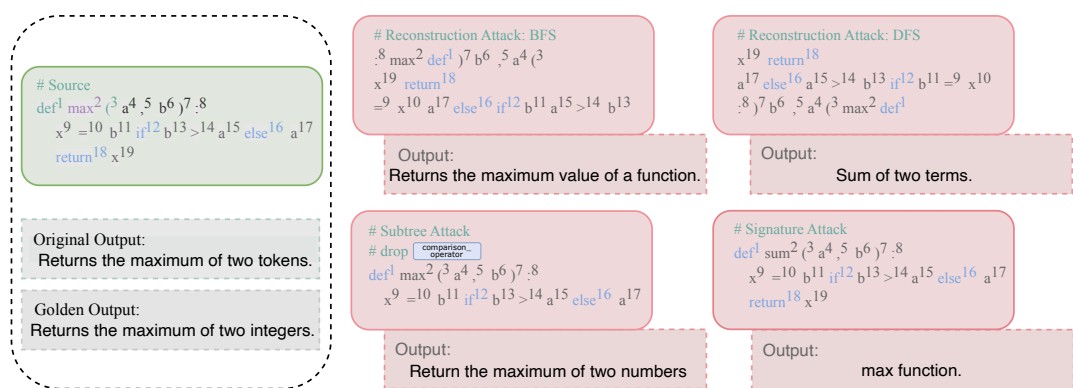

Figure 3: Schematic illustration of four different adversarial sample construction methods, which include strategies based on subtrees of AST, function signatures, and two distinct methods of reconstruction derived from AST traversal. These newly generated samples all exert varying degrees of impact on the model's output (or prediction). However, compared to the original output and golden data, their semantic accuracy is inferior.

## 3.2 Reconstruction

To investigate the sensitivity of CodePTMs to inputs, we considered a framework for introducing perturbations to the code while maintaining an equivalent amount of information. The primary difference between the sequences generated from AST traversal and the original code lies in their structure and order. In the original code, tokens appear in the order they are found in the source code. However, in the sequence generated by traversing the AST using DFS or BFS, the order of the tokens reflects the structure of the code. In a DFS traversal, the order of the tokens is closer to the actual execution order, while in a BFS traversal, the order mainly reflects the hierarchical structure of the code. Furthermore, although the sequence generated by traversing the AST contains all tokens from the original code, it may not fully retain the semantic information of the original code due to the loss of original structural information. For instance, the condition of a conditional statement and its body might be separated, making their relationship less apparent in the sequence.

Previous work has repeatedly demonstrated that code models can capture structural information beneath the textual level (Wan et al., 2022; Chen et al., 2022). Thus, we believe that robust models should be able to extract the necessary information from these "reconstructed" sequences. The procedure is demonstrated in Algorithm 2, where $m$ indicates using BFS or DFS for traversal. The details of AST traversal are shown in Algorithm 3.

## 3.3 Adversarial Training

Following the algorithms proposed in section 3.1, we now introduce adversarial samples during the

---

**Algorithm 2** Reconstruction By Traversal

**Input:** Code snippet $c$, Traversal mode $m$
**Output:** Modified code snippet $c^{'}$

1: **procedure** TRAVERSAL RECONSTRUCTION$(c, m)$
2:     $T \leftarrow \text{GetAST}(c)$
3:     $token\_list \leftarrow \text{list}()$
4:     **for** $node$ in $traverse(c, m)$ **do**
5:         **if** $node$.out_degree $= 0$ and $node$.in_degree $= 1$ **then**
6:             $token\_list$.append($node$.value)
7:         **end if**
8:     **end for**
9:     $c^{'} \leftarrow \text{ListtoCode}(token\_list)$
10:     **return** $c^{'}$
11: **end procedure**

---

training process, enabling the model to be robust enough to make reliable predictions or generate correct sequences when confronted with these intentionally designed perturbations.

We first denote the CodePTM as $M$. Given an input $x$ and corresponding target label or sequence $y$, we define the loss function on original inputs as $L(M, x, y)$ and on adversarial examples as $L_s(M, x, y)$. The construction of adversarial examples, e.g., the use of sub-tree attack to generate samples, is represented by $s$, which transforms $x$ into a new sample while keeping $y$ unchanged. In the procedure of adversarial training, our goal is to find model parameters that minimize the sum of losses on the original inputs and the structured adversarial examples. This objective can be represented as follows:

$$\min_{M}\{L(M, x, y) + L_s(M, x, y)\}\qquad(1)$$

The sum of these two losses signifies the total loss of the model under both normal and adversarial conditions, aiming to enhance the model's robustness against adversarial attacks. To enhance model robustness more effectively, we shuffle the generated adversarial samples randomly and then proceed with training, preventing them from forming dependencies with the original samples during the training process.

## 4 Experiments

### 4.1 Experimental Setup

**Tasks and Datasets**  We conduct our experiments on four tasks related to code representation learning, as part of the CodeXGLUE benchmark (Lu et al., 2021). In the realm of code generation, our first trial is on *code summarization* (Alon et al., 2019), a process aiming to generate natural language comments for a given code snippet. The second one within code generation is *code translation* (Nguyen et al., 2015), which involves translating a code snippet from one programming language to another. For code understanding, we delve into *Clone detection* (Svajlenko et al., 2014; Mou et al., 2016), which quantifies the similarity between different code snippets, and *defect detection* (Zhou et al., 2019), a task focused on predicting the presence of vulnerabilities in the source code, with the potential to pose risks software systems.

**Implementation Details**  In our experiments, we utilized four representative code training models: GraphCodeBERT, PLBART, CodeT5, and UniXcoder. We employed Tree-sitter[1] to parse the source code into ASTs. The training procedure involved using the Adam optimizer (Kingma and Ba, 2015) with a warm-up period of 1,000 steps. Our experiments were conducted using PyTorch 1.5.1 on 4 interconnected NVIDIA RTX 3090 GPUs. The hyperparameters are listed in section B.

### 4.2 Main Results of Attack

**Code Understanding**  As is shown in Table 1, for both clone detection and defect detection tasks, the models exhibit different behaviors under various scenarios. In clone detection, GraphCodeBERT and CodeT5 demonstrate the highest performance

---

[1] github.com/tree-sitter

| Tasks | Clone | Defect | Code Translation | |
|---|---|---|---|---|
| **Language** | Java | C | Java ↔ C# | |
| **Metrics** | F1 | Acc | BLEU | EM |
| *Full Fine-tuning* | | | | |
| GraphCodeBERT | 95.00 | 62.88 | 76.61 | 59.10 |
| PLBART | 93.60 | 63.16 | 80.69 | 64.80 |
| CodeT5 | 95.00 | 65.78 | 81.95 | 66.45 |
| UniXcoder | 91.36 | 62.34 | 76.59 | 63.45 |
| *Reconstruction Attack: DFS* | | | | |
| GraphCodeBERT | 91.61 | 60.29 | 9.33 | 6.25 |
| PLBART | 93.32 | 59.37 | 3.36 | 0.35 |
| CodeT5 | 91.66 | 56.88 | 8.70 | 0.75 |
| UniXcoder | 82.10 | 58.71 | 25.47 | 24.95 |
| *Reconstruction Attack: BFS* | | | | |
| GraphCodeBERT | 91.27 | 59.48 | 15.83 | 15.40 |
| PLBART | 94.12 | 59.41 | 1.76 | 0.10 |
| CodeT5 | 92.37 | 57.39 | 11.94 | 12.95 |
| UniXcoder | 88.50 | 59.37 | 23.84 | 23.75 |
| *Subtree Attack* | | | | |
| GraphCodeBERT | 95.28 | 62.37 | 42.54 | 26.30 |
| PLBART | 93.98 | 62.26 | 10.53 | 6.85 |
| CodeT5 | 95.29 | 62.77 | 47.69 | 31.85 |
| UniXcoder | 89.75 | 60.72 | 37.59 | 25.90 |
| *Signature Attack* | | | | |
| GraphCodeBERT | 95.26 | 62.84 | 73.53 | 57.50 |
| PLBART | 94.42 | 63.29 | 60.05 | 34.15 |
| CodeT5 | 95.10 | 64.34 | 78.08 | 61.80 |
| UniXcoder | 91.32 | 62.36 | 72.26 | 60.75 |

Table 1: Comparative performance of different models on the tasks of clone detection, defect detection, and code translation. Each model's performance is presented under four different structural attacks. For Code Translation, we report the average performance on the tasks of C# - Java and Java - C# translations.

under full fine-tuning. However, when exposed to various attack strategies, all models experience a decrease in performance. Interestingly, PLBART stands out by showing strong robustness, with its performance being particularly resilient under the bfs attack scenario.

From the perspective of attacking strategies, both two reconstruction attacks significantly impact all models in both tasks, indicating the models are hard to understand the sequences with equivalent information in other formats. While under the subtree attack, all models' performance in both tasks is negatively impacted but not as severely as the structural attacks. At last, the Signature attack has variable effects on models. For Clone Detection, GraphCodeBERT manages to maintain a high performance. Notably, for defect detection, PLBART significantly outperforms other models under this

| Languages | Ruby | JavaScript | Go | Python | Java | PHP | Overall |
|---|---|---|---|---|---|---|---|
| *Full Fine-tuning* | | | | | | | |
| GraphCodeBERT | 11.94 | 15.05 | 18.43 | 19.27 | 18.72 | 25.37 | 18.13 |
| PLBART | 14.11 | 15.56 | 18.91 | 19.30 | 18.45 | 23.56 | 18.32 |
| CodeT5 | 15.24 | 16.16 | 19.56 | 20.01 | 20.31 | 26.03 | 19.55 |
| UniXcoder | 14.87 | 15.65 | 19.07 | 19.13 | 20.31 | 26.54 | 19.26 |
| *Reconstruction Attack: DFS* | | | | | | | |
| GraphCodeBERT | 9.35 | 9.54 | 8.97 | 12.43 | 11.61 | 12.47 | 10.73 |
| PLBART | 7.82 | 8.81 | 8.99 | 7.73 | 2.88 | 1.01 | 6.20 |
| CodeT5 | 11.73 | 8.18 | 8.60 | 11.18 | 9.62 | 4.06 | 8.90 |
| UniXcoder | 8.97 | 8.30 | 6.92 | 11.86 | 12.02 | 11.43 | 9.92 |
| *Reconstruction Attack: BFS* | | | | | | | |
| GraphCodeBERT | 10.06 | 11.26 | 11.40 | 14.23 | 11.46 | 12.49 | 11.82 |
| PLBART | 11.03 | 11.37 | 12.53 | 8.70 | 3.87 | 1.06 | 8.09 |
| CodeT5 | 11.73 | 10.55 | 12.92 | 11.64 | 10.23 | 4.07 | 10.19 |
| UniXcoder | 9.97 | 11.15 | 12.77 | 13.55 | 12.67 | 11.47 | 11.93 |
| *Subtree Attack* | | | | | | | |
| GraphCodeBERT | 12.09 | 14.64 | 18.20 | 18.03 | 15.76 | 24.81 | 17.26 |
| PLBART | 13.67 | 15.80 | 18.59 | 18.23 | 9.90 | 20.17 | 16.06 |
| CodeT5 | 14.93 | 15.94 | 19.28 | 18.86 | 17.22 | 25.90 | 18.69 |
| UniXcoder | 14.45 | 15.43 | 19.12 | 18.59 | 16.45 | 25.31 | 18.23 |
| *Signature Attack* | | | | | | | |
| GraphCodeBERT | 9.89 | 10.61 | 12.14 | 15.27 | 18.72 | 14.70 | 13.56 |
| PLBART | 10.77 | 11.25 | 12.24 | 15.43 | 19.17 | 14.26 | 13.85 |
| CodeT5 | 12.13 | 11.91 | 13.44 | 15.84 | 20.34 | 15.36 | 14.84 |
| UniXcoder | 10.24 | 9.70 | 11.50 | 14.73 | 20.19 | 13.10 | 13.24 |

Table 2: Comparative analysis of code summarization performance across various attack strategies. Four representative CodePTMs are evaluated over six programming languages: The performance variation underscores the diverse strengths and weaknesses of these models in the face of different adversarial strategies.

type of attack, achieving an accuracy of 94.42.

**Code Generation** In the task of code translation for Java ↔ C#, CodeT5 performs best under normal fine-tuning scenarios. While under different attack strategies, the performance of all models drops drastically, with UniXcoder often presents the highest performance, especially under both BFS and DFS attacks. This suggests UniXcoder is relatively more robust against these types of structural attacks for the code translation task.

Structural attacks of DFS and BFS cause a drastic decline in performance for all models. Despite this, UniXcoder exhibits relative resilience, managing to maintain the highest performance among all models under these attack scenarios. This implicitly suggests that the training procedure of UniXcoder, which models the AST directly is conducive to model robustness. For the subtree and signature attacks, all models see a decrease in performance, but not as drastic as under the aforementioned DFS/BFS attacks. Nevertheless, CodeT5 consistently outperforms other models under both these attack types. Under the subtree attack, CodeT5 achieves the highest performance, indicating strong robust-

ness. Similarly, for the signature attack, CodeT5 maintains a stable performance that holds the highest scores. These results suggest that CodeT5 may have particular resistance to these types of attacks in the context of code translation, likely due to its pre-training with dataflow information.

In specific cases, it can be observed that the EM metric in the code translation task drops to near zero. Through case studies, we find that this occurs when the model is confronted with a perturbed sequence, its output can be an empty string.

The results of different attack strategies are given in Table 2. Across all programming languages, CodeT5 still consistently outperforms other models in terms of BLEU scores, suggesting its steady performance under various scenarios. Interestingly, the models appear to have difficulties summarizing Ruby code compared to others, which might be attributed to the modest dataset size for this language. On the other hand, summarizing PHP codes is clear to be a task in which all models excel under all kinds of perturbations, likely due to the larger volume of available data, offering models a richer context for capturing the semantics.

Considering the viewpoint of attacking strate-

gies, traversal attacks, namely BFS and DFS, inflict significant damage to the models' performance, with BFS attacks typically causing a more profound impact than DFS. This could be ascribed to the fact that the sequences obtained through DFS have an expression closer to the actual execution order of the code, which aligns more closely with the semantics of the code. Notwithstanding, the models exhibit increased robustness against subtree and signature attacks, maintaining higher performance under these conditions.

## 4.3 Main Results of Adversarial Training

The results of adversarial training are demonstrated in Table 3 and Table 4. For code understanding tasks, it is clear that significant robustness enhancement can be observed for the tasks of clone detection and defect detection. The post-training performance displays considerable resilience and recovery after structural attacks across all CodePTMs. CodeT5, GraphCodeBERT, and PLBART notably improved their performance in both tasks, with CodeT5 generally in the leading position. Although UniXcoder trailed behind in terms of performance, it still exhibited some improvement post-training.

For the cases of enhancing the robustness of models performing the code translation task, all the models demonstrate substantial improvement after the adversarial training. The performance rebound is especially significant in the signature attack scenario where models like CodeT5 reach near fine-tuning results when being exposed to adversarial examples. Moreover, CodePTMs no longer churn out empty strings when faced with perturbed sequences, preventing the previous catastrophic performance decline.

Finally, we can also observe notable improvements after comparing Table 2 and Table 4, showcasing the potential efficacy of adversarial training in enhancing model robustness for the code summarization task. All four CodePTMs exhibit enhancement in robustness, and it is noteworthy that the improvements against the dfs and bfs attacks are significant. These two structural attacks can lead to more severe disruptions to the models, causing larger drops in performance. Therefore, when adversarial training is applied, the resulting improvements can appear more noticeable.

In a nutshell, adversarial training serves as an effective approach for strengthening model robustness and recovery from adversarial attacks. The

| Tasks | Clone | Defect | Code Translation | |
|---|---|---|---|---|
| **Language** | Java | C | Java ↔ C# | |
| **Metrics** | F1 | Acc | BLEU | EM |
| *Full Fine-tuning* | | | | |
| GraphCodeBERT | 95.00 | 62.88 | 76.61 | 59.10 |
| PLBART | 93.60 | 63.16 | 80.69 | 64.80 |
| CodeT5 | 95.00 | 65.78 | 81.95 | 66.45 |
| UniXcoder | 91.36 | 62.34 | 76.59 | 63.45 |
| *Adversarial Training: Reconstruction Attack (DFS)* | | | | |
| GraphCodeBERT | 93.99 | 57.47 | 35.31 | 28.90 |
| PLBART | 93.28 | 59.33 | 27.73 | 29.70 |
| CodeT5 | 94.25 | 54.80 | 46.94 | 38.10 |
| UniXcoder | 65.28 | 58.21 | 46.21 | 37.20 |
| *Adversarial Training: Reconstruction Attack (BFS)* | | | | |
| GraphCodeBERT | 94.07 | 60.61 | 24.76 | 13.45 |
| PLBART | 94.82 | 62.19 | 26.25 | 18.55 |
| CodeT5 | 95.01 | 60.51 | 34.94 | 20.50 |
| UniXcoder | 84.88 | 61.35 | 37.16 | 33.55 |
| *Adversarial Training: Subtree Attack* | | | | |
| GraphCodeBERT | 94.63 | 62.99 | 56.52 | 32.60 |
| PLBART | 94.08 | 61.68 | 55.34 | 30.90 |
| CodeT5 | 95.23 | 63.07 | 66.07 | 39.45 |
| UniXcoder | 88.03 | 61.82 | 59.33 | 32.25 |
| *Adversarial Training: Signature Attack* | | | | |
| GraphCodeBERT | 94.38 | 62.63 | 78.10 | 60.50 |
| PLBART | 94.71 | 63.32 | 81.21 | 63.75 |
| CodeT5 | 94.51 | 63.55 | 82.12 | 66.2 |
| UniXcoder | 90.21 | 62.52 | 76.63 | 63.35 |

Table 3: Results of adversarial training on clone detection, defect detection, and code translation task.

level of improvement, however, varies depending on the model and the specific task. Despite these variations, the trend of enhanced performance post-training remains consistent across all models and tasks, highlighting the value of our method in the realm of programming language models.

## 4.4 Analysis

**Transferability** Adversarial examples constructed through exploiting code structures can effectively enhance the robustness of the model. However, considering that the model may face a variety of adversarial examples concurrently, here we examine its generalization capability. As is demonstrated in Table 5, we evaluate the model trained on adversarial examples constructed via BFS reconstruction on samples reconstructed by the DFS, and vice versa.

**Learning Curve** To validate the necessity of using adversarial examples during the training process, in this part, we set different proportions of adversarial training data to observe how the model

| Languages | Ruby | JavaScript | Go | Python | Java | PHP | Overall |
|---|---|---|---|---|---|---|---|
| *Full Fine-tuning* | | | | | | | |
| GraphCodeBERT | 11.94 | 15.05 | 18.43 | 19.27 | 18.72 | 25.37 | 18.13 |
| PLBART | 14.11 | 15.56 | 18.91 | 19.30 | 18.45 | 23.56 | 18.32 |
| CodeT5 | 15.24 | 16.16 | 19.56 | 20.01 | 20.31 | 26.03 | 19.55 |
| UniXcoder | 14.87 | 15.65 | 19.07 | 19.13 | 20.31 | 26.54 | 19.26 |
| *Adversarial Training: Reconstruction Attack (DFS)* | | | | | | | |
| GraphCodeBERT | 11.54 | 12.98 | 14.91 | 17.18 | 15.78 | 22.28 | 15.78 |
| PLBART | 12.21 | 14.06 | 15.38 | 17.01 | 14.72 | 15.99 | 14.90 |
| CodeT5 | 14.93 | 15.59 | 18.13 | 18.01 | 17.32 | 23.31 | 17.88 |
| UniXcoder | 13.51 | 14.63 | 16.69 | 18.03 | 16.65 | 22.78 | 17.05 |
| *Adversarial Training: Reconstruction Attack (BFS)* | | | | | | | |
| GraphCodeBERT | 10.37 | 10.94 | 12.78 | 14.79 | 14.66 | 19.57 | 13.85 |
| PLBART | 9.42 | 12.41 | 14.48 | 13.46 | 13.07 | 15.09 | 12.99 |
| CodeT5 | 12.90 | 13.87 | 15.62 | 15.42 | 15.79 | 20.45 | 15.68 |
| UniXcoder | 11.34 | 9.57 | 13.33 | 14.91 | 15.46 | 19.53 | 14.02 |
| *Adversarial Training: Subtree Attack* | | | | | | | |
| GraphCodeBERT | 11.77 | 15.23 | 17.81 | 18.02 | 16.26 | 24.59 | 17.28 |
| PLBART | 13.16 | 15.95 | 18.1 | 18.06 | 14.91 | 22.56 | 17.12 |
| CodeT5 | 15.17 | 16.59 | 19.02 | 18.51 | 17.33 | 25.36 | 18.66 |
| UniXcoder | 14.46 | 15.14 | 18.59 | 18.40 | 16.95 | 25.20 | 18.12 |
| *Adversarial Training: Signature Attack* | | | | | | | |
| GraphCodeBERT | 11.38 | 14.54 | 15.76 | 17.68 | 18.67 | 21.68 | 16.62 |
| PLBART | 11.68 | 14.00 | 15.68 | 17.66 | 18.66 | 20.38 | 16.34 |
| CodeT5 | 14.34 | 15.79 | 17.08 | 18.17 | 20.44 | 22.16 | 18.00 |
| UniXcoder | 13.23 | 12.66 | 15.66 | 15.27 | 20.30 | 21.17 | 16.38 |

Table 4: Results of adversarial training on code summarization task.

| Tasks | Clone | Defect | Translation | Summarization |
|---|---|---|---|---|
| Metrics | F1 | Acc | BLEU | BLEU |
| $BFS \rightarrow DFS$ | | | | |
| CodeT5 | 93.45 | 61.02 | 67.55 | 17.94 |
| UniXcoder | 80.99 | 61.31 | 61.07 | 16.60 |
| $DFS \rightarrow BFS$ | | | | |
| CodeT5 | 94.56 | 61.02 | 41.64 | 18.28 |
| UniXcoder | 88.86 | 61.72 | 44.27 | 16.35 |

Table 5: Comparing the generalization capability with adversarial examples generated based on code structure.

learns from the adversarial examples.

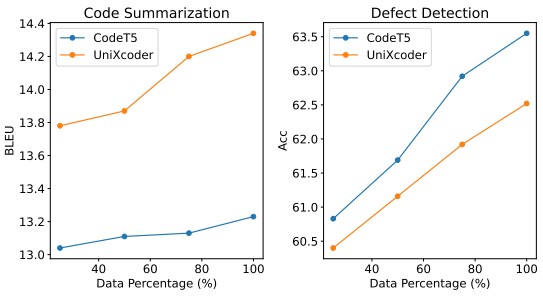

Figure 4: Learning curves of two CodePTMs with different ratios of adversarial examples.

We employ code summarization (Ruby) and defect detection tasks. As clearly shown in Figure 4,

with the increase in the number of adversarial examples, the robustness can be significantly improved for both generative and understanding tasks. This further validates the rationality of our approach to constructing samples based on code structure.

## 5 Conclusion

In this paper, we propose multiple novel attack methods targeting source code from the perspective of code structure. By leveraging the AST of the code, we not only consider constructing adversarial samples that are imperceptible to humans but also create perturbed sequences that preserve the same information as the original samples through their traversal. Then we validate its effectiveness on several mainstream CodePTMs, covering both representative code generation and code understanding tasks. Subsequently, we enhance the model's robustness using adversarial training and investigate the generalizability of performance recovery under different scenarios. Based on our extensive experiments and observations, we provide a comprehensive analysis of the performance of different CodePTMs across various tasks, considering both the vulnerability to attacks, the potential for performance recovery, and the impact on input sensitivity.

## Limitations

- Metrics like BLEU (Papineni et al., 2002; Lin and Och, 2004) and CodeBLEU (Ren et al., 2020) predominantly rely on *n-gram* matching and hence may not adequately consider semantic similarity. Consequently, when evaluating the code sequences generated by models under attack, these metrics could potentially underestimate their semantic correctness.

- Due to the constraints of resources, we confine our backbone models to four representative CodePTMs. While other models (Kanade et al., 2020; Ding et al., 2022) might exhibit slight variance, we hold the view that our current experiments sufficiently encapsulate the most representative scenarios.

## Ethics Statement

The models and data we utilize are all publicly available; our method will not introduce additional model bias and does not involve misuse of code and natural language comments. By designing various attack methods based on code structure, we quantitatively tested the robustness of CodePTMs. Furthermore, we utilize adversarial training to recover the performance of perturbed models and also investigated the correlation between their performance and the number of adversarial samples. We hold the view that these contributions will benefit the NLP research community.

## Acknowledgement

This work has been supported by the National Natural Science Foundation of China under Grant No.U1911203, and the National Natural Science Foundation of China under Grant No.62377012. And the authors would like to thank all the anonymous reviewers for their constructive and insightful comments on this paper.

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

## A Task Overview and Dataset Statistics

### A.1 Defect Detection & Clone Detection

| Dataset | Language | Training | Dev | Testing |
|---|---|---|---|---|
| BigCloneBench | Java | 900K | 416K | 416K |
| Devign | C | 21K | 2.7K | 2.7K |

Table 6: BigCloneBench (Svajlenko et al., 2014) and Devign (Zhou et al., 2019) datasets statistics for Clone detection and Defect Detection tasks.

### A.2 Code Generation & Translation

| Dataset | Language | Training | Dev | Testing |
|---|---|---|---|---|
| CodeTrans | Java - C# | 10,300 | 500 | 1,000 |

Table 7: CONCODE (Iyer et al., 2018) and Code-Trans (Nguyen et al., 2015) datasets statistics for code generation and code translation tasks.

### A.3 Code Summarization

| Language | Training | Dev | Testing |
|---|---|---|---|
| Go | 167,288 | 7,325 | 8,122 |
| Java | 164,923 | 5,183 | 10,955 |
| JavaScript | 58,025 | 3,885 | 3,291 |
| PHP | 241,241 | 12,982 | 14,014 |
| Python | 251,820 | 13,914 | 14,918 |
| Ruby | 24,927 | 1,400 | 1,261 |

Table 8: CodeSearchNet (Husain et al., 2019) data statistics for the code summarization task.

## B Hyperparameters for Fine-tuning

The hyperparameters for tuning CodePTMs by both gold data and adversarial examples are listed in Table 9.

## C Additional Experimental Results

We include additional experimental results in Table 10 and Table 11.

| Hyperparameter | value |
|---|---|
| Batch Size | 8,16,32 |
| Learning Rate | {8e-6, 2e-5, 5e-5} |
| Max Source Length | {130, 240, 256, 320, 512} |
| Max Target Length | {3, 120, 150, 240, 256, 512} |
| Epoch | {2, 30, 50, 120} |

Table 9: Hyperparameters for CodePTM fine-tuning

| Tasks | Clone | Defect | Code Translation | | | |
|---|---|---|---|---|---|---|
| Language | Java | C | Java - C# | | C# - Java | |
| Metrics | F1 | Acc | BLEU | EM | BLEU | EM |
| *Full Fine-tuning* | | | | | | |
| GraphCodeBERT | 95.00 | 62.88 | 80.58 | 59.40 | 72.64 | 58.80 |
| PLBART | 93.60 | 63.16 | 83.02 | 64.60 | 78.35 | 65.00 |
| CodeT5 | 95.00 | 65.78 | 84.03 | 65.90 | 79.87 | 66.90 |
| UniXcoder | 91.36 | 62.34 | 78.95 | 63.30 | 74.22 | 63.60 |
| *Reconstruction Attack: DFS* | | | | | | |
| GraphCodeBERT | 91.61 | 60.29 | 10.18 | 0.10 | 8.47 | 12.40 |
| PLBART | 93.32 | 59.37 | 6.52 | 0.70 | 0.20 | 0.00 |
| CodeT5 | 91.66 | 56.88 | 15.38 | 0.60 | 2.02 | 0.90 |
| UniXcoder | 82.10 | 58.71 | 32.32 | 24.40 | 18.62 | 25.50 |
| *Reconstruction Attack: BFS* | | | | | | |
| GraphCodeBERT | 91.27 | 59.48 | 24.39 | 8.80 | 7.27 | 22.00 |
| PLBART | 94.12 | 59.41 | 3.43 | 0.20 | 0.09 | 0.00 |
| CodeT5 | 92.37 | 57.39 | 17.97 | 14.90 | 5.91 | 11.00 |
| UniXcoder | 88.50 | 59.37 | 32.07 | 25.10 | 15.60 | 22.40 |
| *Subtree Attack* | | | | | | |
| GraphCodeBERT | 95.28 | 62.37 | 50.13 | 24.90 | 34.95 | 27.70 |
| PLBART | 93.98 | 62.26 | 19.81 | 13.60 | 1.24 | 0.10 |
| CodeT5 | 95.29 | 62.77 | 63.20 | 33.20 | 32.17 | 30.50 |
| UniXcoder | 89.75 | 60.72 | 54.68 | 30.10 | 20.50 | 21.70 |
| *Signature Attack* | | | | | | |
| GraphCodeBERT | 95.26 | 62.84 | 76.10 | 56.80 | 70.96 | 58.20 |
| PLBART | 63.29 | 94.42 | 64.73 | 33.90 | 55.37 | 34.40 |
| CodeT5 | 64.34 | 95.10 | 80.92 | 62.70 | 75.23 | 60.90 |
| UniXcoder | 91.32 | 62.36 | 76.92 | 60.40 | 67.59 | 61.10 |

Table 10: Comparative performance of different models on the tasks of clone detection, defect detection, and code translation. Each model's performance is presented under four different structural attacks.

## D Detailed Algorithm for Reconstruction Attack

Due to the space constraints, we put a simplified version of the reconstruction attack in section 3.2. Algorithm 3 is a more detailed version.

## E Case Studies

To better understand the effect of adversarial samples generated through exploiting code structure in specific tasks, we present a series of case studies in Table 12, Table 13, and Table 14 for subtree attack, signature attack, and reconstruction attack respectively.

| Tasks | Clone | Defect | Code Translation | | | |
|---|---|---|---|---|---|---|
| **Language** | Java | C | Java - C# | | C# - Java | |
| **Metrics** | F1 | Acc | BLEU | EM | BLEU | EM |
| *Full Fine-tuning* | | | | | | |
| PLBART | 93.60 | 63.16 | 83.02 | 64.60 | 78.35 | 65.00 |
| CodeT5 | 95.00 | 65.78 | 84.03 | 65.90 | 79.87 | 66.90 |
| GraphCodeBERT | 95.00 | 62.88 | 80.58 | 59.40 | 72.64 | 58.80 |
| UniXcoder | 91.36 | 62.34 | 78.95 | 63.30 | 74.22 | 63.60 |
| *Adversarial Training: Reconstruction Attack (DFS)* | | | | | | |
| PLBART | 93.28 | 59.33 | 34.20 | 31.70 | 21.26 | 27.70 |
| CodeT5 | 94.25 | 54.80 | 54.88 | 42.00 | 39.00 | 34.20 |
| GraphCodeBERT | 93.99 | 57.47 | 43.20 | 30.6 | 27.41 | 27.20 |
| UniXcoder | 65.28 | 58.21 | 55.37 | 41.00 | 37.05 | 33.40 |
| *Adversarial Training: Reconstruction Attack (BFS)* | | | | | | |
| PLBART | 94.82 | 62.19 | 42.89 | 34.80 | 9.60 | 2.30 |
| CodeT5 | 95.01 | 60.51 | 49.04 | 36.90 | 20.84 | 4.10 |
| GraphCodeBERT | 94.07 | 60.61 | 39.49 | 25.90 | 10.02 | 1.00 |
| UniXcoder | 84.88 | 61.35 | 46.25 | 37.80 | 28.06 | 29.30 |
| *Adversarial Training: Subtree Attack* | | | | | | |
| PLBART | 94.08 | 61.68 | 55.83 | 30.8 | 54.84 | 31.00 |
| CodeT5 | 95.23 | 63.07 | 71.41 | 39.60 | 60.73 | 39.30 |
| GraphCodeBERT | 94.63 | 62.99 | 63.36 | 31.20 | 51.87 | 34.00 |
| UniXcoder | 88.03 | 61.82 | 63.36 | 34.30 | 55.29 | 30.20 |
| *Adversarial Training: Signature Attack* | | | | | | |
| PLBART | 94.71 | 63.32 | 82.34 | 61.00 | 80.07 | 66.50 |
| CodeT5 | 94.51 | 62.55 | 84.77 | 66.7 | 79.46 | 65.70 |
| GraphCodeBERT | 94.38 | 62.63 | 80.61 | 60.20 | 75.58 | 60.80 |
| UniXcoder | 90.21 | 62.52 | 78.78 | 63.60 | 74.48 | 63.10 |

Table 11: Comparative performance of adversarial training of different models on the tasks of clone detection, defect detection, and code translation.

---

**Algorithm 3** Traverse Tree

**Input:** Code snippet $c$, Traversal mode $m$
**Output:** Modified code snippet $c^{'}$

1: **procedure** TRAVERSAL($c, m$)
2:     $node\_type \leftarrow \text{list}()$
3:     $T \leftarrow \text{GetAST}(c)$
4:     $queue \leftarrow \text{list}()$
5:     $queue.\text{append}(tree.root)$
6:     **while** queue **do**
7:         **if** $m$ = "DFS" **then**
8:             $current\_node \leftarrow \text{queue.pop}()$
9:         **else if** $m$ = "BFS" **then**
10:        $current\_node \leftarrow \text{queue.pop}(0)$
11:       **end if**
12:     $node\_type.\text{append}(current\_node.type)$
13:     **if** $m$ = "DFS" **then**
14:        **for** $child \in$ $current\_node.children[::-1]$ **do**
15:          queue.append($child$)
16:        **end for**
17:     **else if** $m$ = "BFS" **then**
18:        **for** $child \in current\_node.children$ **do**
19:          queue.append($child$)
20:        **end for**
21:     **end if**
22:     **end while**
23:     **return** $queue$
24: **end procedure**

| **Original** | **After Attack** |
|---|---|

```
1 public NotImplementedFunctionException(
     String functionName,
     NotImplementedException cause) {
2   super(functionName, cause);
3   this.functionName = functionName;
4 }
```

Code 1: Original input.

```
1 NotImplementedFunctionException ( String
     functionName ,
     NotImplementedException cause ) {
2   super ( functionName , cause ) ;
3   = functionName ;
4 }
```

Code 2: Input under subtree attack.

```
1 NotImplementedFunctionException ( String
     functionName ,
     NotImplementedException cause ) {
2   super ( functionName , cause ) ;
3   = functionName ;
4 }
```

Code 3: Generated codes based on original input.

```
1 FunctionException(string functionName,
     NotImplementedException cause)
2   : base(functionName, cause)
3 {
4     _functionName = functionName;
5 }
```

Code 4: Generated codes based on input under subtree attack.

Table 12: Case studies of code translation under subtree attack.

| **Original** | **After Attack** |
|---|---|

```
1 def dailymotion_download(url, output_dir=
     '.', merge=True, info_only=False, **
     kwargs):
2   html = get_content(rebuilt_url(url))
3   info = json.loads(match1(html, r'
       qualities":({.+?}),"'))
4   title = match1(html, r'"video_title"\
       s*:\s*"([^"]+)"') or match1(html,
       r'"title"\s*:\s*"([^"]+)"')
5   title = unicodize(title)
6
7   for quality in ['1080', '720', '480',
       '380', '240', '144', 'auto']:
8     try:
9         real_url = info[quality][1]["
             url"]
10        if real_url:
11            break
12    except KeyError:
13        pass
14
15  mime, ext, size = url_info(real_url)
16  print_info(site_info, title, mime,
       size)
17
18  if not info_only:
19    download_urls([real_url], title,
         ext, size, output_dir=
         output_dir, merge=merge)
```

Code 5: Original input.

```
1 def sum (url, output_dir='.', merge=True,
     info_only=False, **kwargs):
2   html = get_content(rebuilt_url(url))
3   info = json.loads(match1(html, r'
       qualities":({.+?}),"'))
4   title = match1(html, r'"video_title"\
       s*:\s*"([^"]+)"') or match1(html,
       r'"title"\s*:\s*"([^"]+)"')
5   title = unicodize(title)
6
7   for quality in ['1080', '720', '480',
       '380', '240', '144', 'auto']:
8     try:
9         real_url = info[quality][1]["
             url"]
10        if real_url:
11            break
12    except KeyError:
13        pass
14
15  mime, ext, size = url_info(real_url)
16  print_info(site_info, title, mime,
       size)
17
18  if not info_only:
19    download_urls([real_url], title,
         ext, size, output_dir=
         output_dir, merge=merge)
```

Code 6: Input under signature attack.

```
1 # Download a website .
```

Code 7: The summarization for original code.

```
1 # Summarize a URL .
```

Code 8: The summarization for the attacked code.

Table 13: Case studies of code summarization on Python under signature attack.

| **Original** | **After Attack** |
|---|---|

```
1 def wix_light_extension(extension)
2   unless extension.is_a?(String)
3     raise InvalidValue.new(:
          wix_light_extension, "be␣an␣
          String")
4   end
5   wix_light_extensions << extension
6 end
```

Code 9: Original input.

```
1 end
2   wix_light_extension def ) extension (
3   extension << wix_light_extensions
4   end
5   unless is_a? . extension raise) String
        (
6     new . InvalidValue ) , :
          wix_light_extension ( "␣be␣an␣
          String␣"
```

Code 10: Input under reconstruction attack.

```
1 # Adds a Wix Light Extension .
```

```
1 # wix_light_extension
```

Code 11: The summarization for original code.

Code 12: The summarization for the attacked code.

Table 14: Case studies of code summarization on Ruby under reconstruction attack (BFS).

| Original Input1 | Original Input2 |
|---|---|

```
1  private void download(String address,
       String localFileName) throws
       UrlNotFoundException, Exception {
2      String ext = G_File.getExtensao(
           address);
3      if (ext.equals("jsp")) {
4          throw new Exception("Erro␣ao␣
               baixar␣pagina␣JSP,␣tipo␣
               negado." + address);
5      }
6      File temp = new File(localFileName +
           ".tmp");
7      if (temp.exists()) temp.delete();
8      OutputStream out = null;
9      URLConnection conn = null;
10     InputStream in = null;
11     try {
12         try {
13             URL url = new URL(address);
14             conn = url.openConnection();
15             in = conn.getInputStream();
16         } catch (FileNotFoundException e2)
               {
17             throw new UrlNotFoundException
                   ();
18         }
19         out = new BufferedOutputStream(
               new FileOutputStream(temp));
20         byte[] buffer = new byte[1024];
21         int numRead;
22         long numWritten = 0;
23         while ((numRead = in.read(buffer))
               != -1) {
24             out.write(buffer, 0, numRead);
25             numWritten += numRead;
26         }
27     } catch (UrlNotFoundException
           exception) {
28         throw exception;
29     } catch (Exception exception) {
30         throw exception;
31     } finally {
32         try {
33             if (in != null) {
34                 in.close();
35             }
36             if (out != null) {
37                 out.close();
38             }
39         } catch (IOException ioe) {
40         }
41     }
42 }
```

```
1  private static void copyFile(File src,
       File dst) throws IOException {
2      FileChannel in = new FileInputStream(
           src).getChannel();
3      FileChannel out = new
           FileOutputStream(dst).getChannel
           ();
4      in.transferTo(0, in.size(), out);
5      in.close();
6      out.close();
7  }
```

Code 14: Original input.

Code 13: Original input.

Table 15: Case studies of clone detection under reconstruction attack (DFS), where model can accurately judge the relation between two code snippets.

**After Attack**

```
 1  }
 2      }
 3          }
 4          { ) ioe IOException ( catch }
 5              }
 6                  ; ) ( close . out
 7              { ) null != out ( if
 8              }
 9                  ; ) ( close . in
10              { ) null != in ( if
11          { try
12  { finally }
13      ; exception throw
14  { ) Exception ( catch }
15      ; exception throw
16  { ) UrlNotFoundException ( catch }
17          }
18          numRead += numWritten ;
19          ) numRead , 0 , buffer ( write . out
20      { ) 1 - != ) ) buffer ( read . in = numRead ( ( while
21      ; 0 = numWritten long
22      ; numRead int
23      ; ] 1024 [ byte new = buffer ] [ byte
24      ; ) ) temp ( FileOutputStream ( BufferedOutputStream
              new = out
25          }
26          ) ( UrlNotFoundException new throw
27      { ) FileNotFoundException ( catch }
28          ; ) ( getInputStream . conn = in
29          ; ) ( openConnection . url = conn
30          ; ) address ( URL new = url URL
31      { try
32  { try
33  ; null = in InputStream
34  ; null = conn URLConnection
35  ; null = out OutputStream
36  ; ) ( delete . ) ) ( exists . temp ( if
37  ) "␣" + localFileName ( File new = temp File
38  }
39      ) address + "Erro␣ao␣baixar␣pagina␣JSP,␣tipo␣negado."
            ( Exception new throw
40  { ) ) "␣" ( equals . ext ( if
41  ; ) address ( getExtensao . G_File = ext String
42  { , UrlNotFoundException throws ) localFileName , address (
      download void private
```

Code 15: Input1 under reconstruction attack (DFS)

Table 16: Case studies of clone detection under reconstruction attack (DFS). The prediction of the model is altered after the attack, generating a wrong prediction for semantic similarity.