# OpenReview forum: "Evaluating and Enhancing the Robustness of Code Pre-trained Models through Structure-Aware Adversarial Samples Generation"
_EMNLP/2023/Conference — EMNLP 2023 Findings_

### Official Review · Reviewer_xnBH · 2023-07-29

**Soundness:** 2

**Excitement:**

3: Ambivalent: It has merits (e.g., it reports state-of-the-art results, the idea is nice), but there are key weaknesses (e.g., it describes incremental work), and it can significantly benefit from another round of revision. However, I won't object to accepting it if my co-reviewers champion it.

**Paper Topic And Main Contributions:**

This paper studies code models' resilience to adversarial attacks. The authors propoe a series of new attacks that consider code structure, noting current attacks overlook this aspect. They evaluate these attacks on multiple code models and tasks, demonstrating that adversarial training strengthens model robustness against these attacks.

**Reasons To Accept:**

- The authors propose several novel attacks that consider the structure of the code

- The authors present an extensive evaluation of the robustness of code models to adversarial attacks, considering various code models and associated downstream tasks.


**Reasons To Reject:**

- The authors state that their work is motivated by a disregard for code structure in existing methods of generating adversarial code attacks. However, the necessity and advantage of considering code structure is not explicitly articulated.

- In the experiments there are no comparisons against current approaches, so it remains unclear whether the proposed attacks offer any advantages.

- The manuscript lacks clarity and does not have proper references to some figures and appendix

**Reproducibility:**

4: Could mostly reproduce the results, but there may be some variation because of sample variance or minor variations in their interpretation of the protocol or method.

**Reviewer Confidence:**

4: Quite sure. I tried to check the important points carefully. It's unlikely, though conceivable, that I missed something that should affect my ratings.

**Typos Grammar Style And Presentation Improvements:**

- It is not clear what the authors would like to achieve with the reconstruction task

- Figure 2 and Figure 3 are not referenced in the main text

- The material in the appendix lacks referencing in the main body of the text

---

> ### Author Rebuttal · Authors · 2023-08-28
>
> Thanks for reviewing our paper, we hope that our answers can address your concerns.
>
> > R1: The authors state that their work is motivated by a disregard for code structure in existing methods of generating adversarial code attacks. However, the necessity and advantage of considering code structure are not explicitly articulated.
>
> Leveraging code structure to assist downstream task learning is relatively common in the neural code intelligence field. In this paper, we utilize one such form, namely the AST (Abstract Syntax Tree), to construct adversarial code attack samples. Without considering structure, one can only construct samples from a textual perspective. Such samples would merely impact the model's discriminative performance at the textual level.
>
> > R2: In the experiments, there are no comparisons against current approaches, so it remains unclear whether the proposed attacks offer any advantages.
>
> We chose to compare with full fine-tuning to better explore the performance difference before and after the attacks. This matches better for our case that the attack is conducted at the sequence level. Other methods like BERT-ATTACK[1] don't exploit code structure for attacks, and the attack is conducted under token level, while ours mainly relies on sequence level. Additionally, one code-related baseline CodeAttack[2] uses "a small modification" to construct adversarial samples that also do not match our cases, since the rate of "tokens" being modified varies too much.
>
> [1] BERT-ATTACK: Adversarial Attack Against BERT Using BERT EMNLP 2020
>
> [2] CodeAttack: Code-Based Adversarial Attacks for Pre-trained Programming Language Models AAAI 2023
>
> > R3: The manuscript lacks clarity and does not have proper references to some figures and appendix
>
> Thank you for raising concerns about the quality of the manuscript. Due to space constraints, we had to condense some descriptions and details. In subsequent revisions, we will strive to address these issues and, if necessary, add additional descriptions to the appendices.
>
> > Typos Grammar Style And Presentation Improvements
>
> We deeply appreciate your meticulous review of our paper. The reconstruction task is specifically designed for the structure of the code, using different traversal methods to measure the model's robustness while retaining most of the code information. Additionally, we thank you for pointing out the oversight regarding Figure 2 and Figure 3 not being referenced in the main text. We will address this in our next revisions.

---

### Official Review · Reviewer_BNiS · 2023-08-04

**Soundness:** 3

**Excitement:**

4: Strong: This paper deepens the understanding of some phenomenon or lowers the barriers to an existing research direction.

**Missing References:**

Did the authors conduct experiments to check the compilation rate of the adversarial code (both before and after the attack)? And as a result if the compilation rate of the generated code is impacted in case of code generation tasks?


**Paper Topic And Main Contributions:**

The authors propose a set of novel evaluation techniques for attacking pre-trained models of code. They generate adversarial samples that are imperceptible to humans by attacking identifier tokens, sub-trees and use AST structures to generate an adversarial attack.

**Reasons To Accept:**

+ The generated adversarial code is imperceptible to humans.
+ Generate adversarial samples that attack identifier tokens and sub-tree structures.
+ Extensive experiments on code understanding and code generation tasks.

**Reasons To Reject:**

- Missing comparison with other baseline models that attack pre-trained models of code.

**Reproducibility:**

3: Could reproduce the results with some difficulty. The settings of parameters are underspecified or subjectively determined; the training/evaluation data are not widely available.

**Reviewer Confidence:**

3: Pretty sure, but there's a chance I missed something. Although I have a good feel for this area in general, I did not carefully check the paper's details, e.g., the math, experimental design, or novelty.

---

> ### Author Rebuttal · Authors · 2023-08-28
>
> Thanks for providing your review, and hope our responses can address your concerns:
>
> > Missing comparison with other baseline models that attack pre-trained models of code.
>
> Thank you for raising concerns regarding the baselines. Our work covers both code generation and code understanding scenarios and introduces four different strategies for attacks and adversarial training. Finding similar baselines can be challenging due to this scope. Specifically, methods like BERT-ATTACK[1] don't exploit code structure for attacks, and the attack is conducted under token level, while ours mainly relies on sequence level. Moreover, one recent code-related baseline CodeAttack[2] uses "a small modification" to create adversarial samples that also do not match our cases.
>
> [1] BERT-ATTACK: Adversarial Attack Against BERT Using BERT EMNLP 2020
>
> [2] CodeAttack: Code-Based Adversarial Attacks for Pre-trained Programming Language Models AAAI 2023
>
> > Questions in "Missing References"
>
> We interpret the "compilation rate" you mentioned as the ratio of two types of samples. In the specific adversarial training process, we used an equal number of samples. For code generation tasks, there will indeed be an impact, as both our attack and the reconstruction method are based on structural information that changes the original form of the code. We appreciate your query and will clarify these details in the next revisions.

---

### Official Review · Reviewer_H95H · 2023-08-05

**Soundness:** 3

**Excitement:**

4: Strong: This paper deepens the understanding of some phenomenon or lowers the barriers to an existing research direction.

**Paper Topic And Main Contributions:**

This paper proposes a new attack method to attack the models in field of neural coding intelligence and reveals the vulnerabilities of such models. Specifically, the author proposes several attack methods based on the structured of the code. These methods mainly use the signature of function, abstract syntax trees, and logic structure for input perturbation. By the way, the author shows the performance of purposed methods through experiments. In addition, the author demonstrated the transferability of the proposed method and the necessity of relevant adversarial training through experiments .

**Questions For The Authors:**

Question A:
What are the experimental results of the unstructured method？

Question B:
Can samples obtained through significant changes in sample structure and content still be called adversarial samples?

Question C:
Why is code summarization classified as a task in the field of code generation?

Question D:
What is the composition of the data used for adversarial training?

Question E:
The adversarial training enhances the robustness of the model when facing corresponding attacks, but will there be any changes in performance when dealing with other attacks?

**Reasons To Accept:**

1. The proposed method is intuitive.

2. The experimental results are promising.

**Reasons To Reject:**

1.	Lack of comparative experiments with traditional methods.

The paper conducted comparative experiments and analysis on the proposed structured adversarial attack methods, but did not compare them with other unstructured adversarial attack methods, which cannot effectively reflect the specificity and effectiveness of structured information.


2.	This work may have soundness issues.

(1)	As mentioned in Section 4.1, code summarization is a process aiming to generate natural language comments for a given code snippet. This undertaking necessitates the model to comprehend the input code and generate an appropriate natural language summary. It is important to note that this process does not involve code generation per se, but rather leans towards code comprehension. Nevertheless, the authors classify it within the domain of code generation, which is unreasonable.

(2)	Some of the conclusions do not match the experimental results provided. The authors claimed that “Notably, for defect detection, PLBART significantly outperforms other models under this type of attack, achieving an accuracy of 94.42.” However, Table 1 shows that the accuracy of the PLBART model in this attack is only 63.29, which is not consistent with the 94.42 mentioned in the paper and does not have an advantage compared to other models.

(3)	Adversarial samples are usually obtained by adding small perturbations. Can samples obtained through significant changes in sample structure and content still be called adversarial samples? The adversarial samples produced by the Reconstruction Attack and Subtree Attack will significantly modify the source code, leading to numerous conspicuous flaws in the generated code. So, when dealing with tasks such as code summarization and defect detection, is it reasonable to expect the model to return Golden Output?

3.	Important details are missing.
The authors conducted adversarial training on the models, but many important details of the training were not provided. For example, what is the composition of the data used for adversarial training?

4.	The adversarial training enhances the robustness of the model when facing corresponding attacks, but will there be any changes in performance when dealing with other attacks?

**Reproducibility:**

3: Could reproduce the results with some difficulty. The settings of parameters are underspecified or subjectively determined; the training/evaluation data are not widely available.

**Reviewer Confidence:**

3: Pretty sure, but there's a chance I missed something. Although I have a good feel for this area in general, I did not carefully check the paper's details, e.g., the math, experimental design, or novelty.

---

> ### Author Rebuttal · Authors · 2023-08-28
>
> Thanks a lot for your insightful comments, and we hope our rebuttal can address your concerns.
>
> > Question A: What are the experimental results of the unstructured method？
>
> Thank you for your question and the feedback provided in the "reason to reject" section. Our work covers both code generation and code understanding scenarios and proposes four different strategies for sequence-level attack and adversarial training, making it challenging to find a comparable baseline (e.g., previous attacks mainly focus on token-level substitution). Specifically, methods in NLP like BERT-ATTACK[1] don't utilize code structure / syntactic information for attacks, and one recent code-related baseline CodeAttack[2] uses "a small modification" to create adversarial samples that also do not match our scenario.
>
> [1] BERT-ATTACK: Adversarial Attack Against BERT Using BERT EMNLP 2020
>
> [2]  CodeAttack: Code-Based Adversarial Attacks for Pre-trained Programming Language Models AAAI 2023
>
> > Question B: Can samples obtained through significant changes in sample structure and content still be called adversarial samples?
>
> Yes, we believe so. In terms of reconstruction, the samples we constructed using methods like DFS and BFS retain most of the original structural information of the code. Previous works, as referenced in Line 242-244, have demonstrated that CodePTMs can capture such information.
>
> > Question C: Why is code summarization classified as a task in the field of code generation?
>
> The code summarization task is sourced from the CodeXGLUE[1] Benchmark, specifically under the Code2NL task. This task involves writing comments for given code snippets, so it is classified into code generation task in [1]. We will clarify this issue in the next version.
>
> [1] CodeXGLUE: A Machine Learning Benchmark Dataset for Code Understanding and Generation
>
> > Question D: What is the composition of the data used for adversarial training?
>
> Thank you for your interest in our experimental design. In the specific adversarial training process, we used an equal number of two types of samples, namely the original code and the reconstructed code after our adversarial modifications.
>
> > Question E: The adversarial training enhances the robustness of the model when facing corresponding attacks, but will there be any changes in performance when dealing with other attacks?
>
> Thank you for raising the question regarding the strategy of attacks. We have addressed the generalization of different attacks in the analysis section, specifically in Figure 4. We will try to further analyze the changes when dealing with other attacks.
>
> > Responses to Reasons To Reject
>
> Thank you for your thorough review and comments. We acknowledge that due to page constraints, certain details were unfortunately omitted, and we sincerely apologize for any confusion caused. With respect to the PLBART conclusion, we regret the oversight, and it pertains to clone detection. The accuracy for clone detection in PLBART is 94.42 (as indicated in the third-to-last row), while the accuracy for defect detection is 63.29. Corrections will be made accordingly in the next revision.
>
> For other points you've raised, please refer to our earlier responses to address your questions. We will ensure that more comprehensive experimental details are included in our next version to enhance the clarity and quality of the paper.
>
> Once again, thank you for your valuable feedback and efforts towards improving our manuscript.

---

### Meta-Review · Area_Chair_pazq · 2023-09-19

**Recommendation:** 3

**Metareview:**

The reviewers were enthusiastic about the paper and agreed that the methodology was intuitive and sufficiently novel, and the presentation of the results were clear. The rebuttal phase helps clarify some of the confusion and answer outstanding questions. As one reviewer pointed out, the paper may need additional work to situate it against existing work and choosing appropriate comparators. The authors response indicates that this merits some discussion in the revised version. There is strong interest in this line of work and the excitement on the proposed approaches is high.

---

### Decision · Program_Chairs · 2023-10-07

**Decision:**

Accept-Findings

**Comment:**

The reviewers were enthusiastic about the paper and agreed that the methodology was intuitive and sufficiently novel, and the presentation of the results were clear. The rebuttal phase helps clarify some of the confusion and answer outstanding questions. As one reviewer pointed out, the paper may need additional work to situate it against existing work and choosing appropriate comparators. The authors response indicates that this merits some discussion in the revised version. There is strong interest in this line of work and the excitement on the proposed approaches is high.